# End-of-Life Textile Recognition in a Circular Economy Perspective: A Methodological Approach Based on Near Infrared Spectroscopy

Giuseppe Bonifazi [1,2], Riccardo Gasbarrone [1], Roberta Palmieri [1] and Silvia Serranti [1,2,*]

[1] Department of Chemical Engineering, Materials and Environment, Sapienza-University of Rome, Via Eudossiana 18, 00184 Rome, Italy
[2] Research Center for Biophotonics, Sapienza-University of Rome, Polo Pontino, Corso della Repubblica 79, 04100 Latina, Italy
* Correspondence: silvia.serranti@uniroma1.it

**Abstract:** The life cycle of textiles (i.e., fabrics and apparel products) generates many environmental impacts, such as resource consumption, water, soil, and air pollution through the dispersion of chemical substances and greenhouse gases. For these reasons, in 2019, textiles were identified as a "priority product category for the circular economy" by the European Commission that proposed a new circular economy action plan focusing on recycling. An in-depth characterization of textile fabrics could lead to an ad hoc recycling procedure, reducing resource consumption and chemicals utilization. In this work, NIR (1000–1650 nm) spectroscopy was applied to extract information regarding fabric composition, with reference to cotton, silk, viscose, and some of their blends, using two different devices: a hyperspectral imaging (HSI) platform and a portable spectroradiometer. The different fabrics were correctly classified based on their spectral features by both detection instruments. The proposed methodological approach can be applied for quality control in the textile recycling sector at industrial and/or laboratory scale thanks to the easiness of use and the speed of detection.

**Keywords:** end-of-life textiles; fabric; waste characterization; hyperspectral imaging; recycling; circular economy; near infrared spectroscopy

## 1. Introduction

Textiles are, after food, housing, and transport, the fourth highest pressure category for primary raw materials and water use, the second highest for land use, and the fifth for greenhouse gas (GHG) emissions [1]. Indeed, according to the United Nations website, the fashion industry (i.e., clothing and footwear) produces more than 8% of the global greenhouse gases and 20% of the wastewater every year [2]. Moreover, this industry consumes a large amount of non-renewable resources and requires treatment processes frequently employing polluting and hazardous substances [3]. It is estimated that the textile industry is responsible for 10% of the global carbon emissions and it will consume up to 26% of the world carbon budget by 2050 [4–6]. A total of 150 million tons of textile waste is generated all over the world every year, 5.8 of which is discarded by European consumers, corresponding to 11.3 kg per person [7]. Therefore, addressing the negative effects of the textile industry through the benefits of circular economy strategies is of primary importance for sustainability purposes [8]. Reuse and recycling are the most effective methods of textile waste disposal, with the lowest environmental impacts. However, currently less than 1% of all textiles are recycled into new textiles globally [9]. More in detail, the textile recycling rate is about 25% in Europe and even lower in the United States, about 16.2% [10], indicating that instead of being collected and recycled, millions of tons of textiles end up in landfills. Considering that almost all textiles are fully recyclable, landfilling should be the

last management option for this waste [11]. Increasing textile recycling rates would reduce the already mentioned negative environmental impacts linked to this industrial sector [12].

In this scenario, the European Commission has identified textiles (i.e., apparel and fabrics) as a "priority product category for the circular economy", encouraging the development of technological applications for textile waste reuse and recycling. Moreover, the European Directive EU 2018/851 obliges Member States to start the separate collection of textile waste by 1 January 2025.

In this framework, textile recycling technologies are rapidly growing, with the aim of increasing the quantity and quality of produced secondary raw materials. One of the most important issues in textile recycling is the identification of the fiber typologies constituting the waste. From a recycling perspective, textile fiber identification and sorting according to material/blend is of primary importance to enable the implementation of a correct recycling system [13].

Textile fibers are usually classified as natural or man-made [14]. In more detail, natural fibers can be of: (i) vegetable-based origin (i.e., cellulosic fibers obtained from different parts of plants such as leaves and seeds), (ii) animal-based origin (i.e., protein fibers), and (iii) mineral-based origin (i.e., glass fibers). Man-made fibers are artificial materials that can be regenerated (i.e., viscose derived from cellulose), or synthetic (mostly fossil-based). Textile waste requires different recycling treatments based on their composition. The application of low-cost and automatic systems for the identification and sorting of end-of-life textiles plays an important role in the feasibility of an efficient recycling process [15]. Currently, waste textile sorting takes place mainly through manual operations, accounting for 30% of the entire recycled textile cost [16]. The realization of an automatic sorting system could dramatically reduce process costs and time.

The present work is addressed to explore the use near infrared (NIR) spectroscopy for the recognition of end-of-life natural textile fibers, such as cotton (i.e., vegetable-based origin) and silk (i.e., animal-based origin), and artificial ones, as viscose, and some of their blends. In more detail, a methodological approach based on the utilization of two different devices working in the NIR range (1000–1650 nm), a hyperspectral imaging (HSI) system, and a portable single-spot spectroradiometer, were developed, set up, and tested for textile fiber sorting and/or quality control in a recycling process.

NIR spectroscopy and chemometric analyses are widely utilized for material characterization, classification, and quality control in several sectors, such as primary/secondary raw materials [17–19], pharmaceutical and chemical industry [20–22], cultural heritage, agricultural/food industry [23–26], medicine and clinical applications [27,28], and, more generally, in analytical science [29] to perform systematic environmental remote and proximal sensing.

## 2. Materials and Methods

### 2.1. Analyzed Samples

Five textile samples of different colors and compositions have been selected from end-of-life (EoL) apparels (Figure 1). In more detail, the type of fabric was identified according to the composition reported by the producers on the apparel labels as:

(a) 100% Cotton;
(b) 100% Silk;
(c) 100% Viscose;
(d) 20% Cotton–80% Viscose;
(e) 50% Cotton–50% Silk.

Cotton and silk are both natural fibers, the first vegetable-based and the second animal-based. On the contrary, viscose is usually made of wood cellulose and synthetic substances.

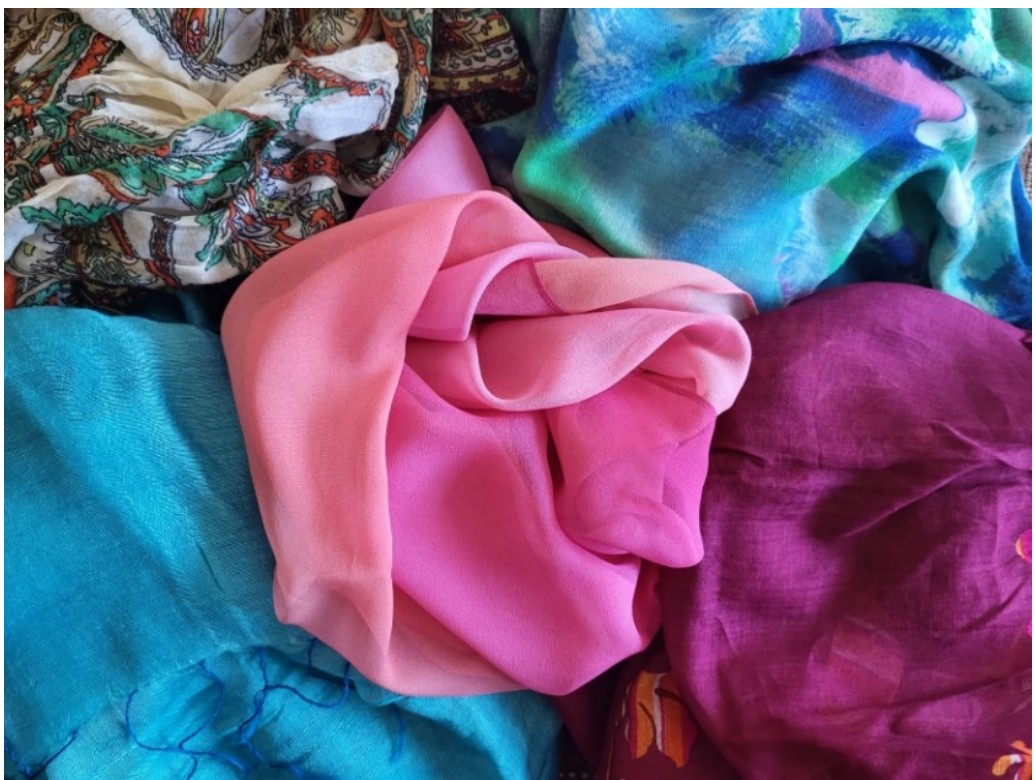

**Figure 1.** Selected end-of-life textile apparel.

*2.2. Methods*

2.2.1. Hyperspectral Imaging System

A HSI system is essentially based on an integrated hardware and software architecture allowing one to acquire and handle spectral data as an image sequence, resulting from a pre-defined alignment on a sample surface properly energized by a light source [30,31].

The collected spectral information generates a three-dimensional dataset, the so-called "hypercube", characterized by two spatial dimensions (X, Y) and one spectral dimension (λ). The 3D nature of a hyperspectral image enables the investigation of several physical–chemical characteristics of a sample surface based on the different collected wavelengths and selected instruments.

The acquisition of hyperspectral images was performed by a NIR Spectral Camera$^{TM}$ equipped with an ImSpector$^{TM}$ N17E (SPECIM Ltd., Oulu, Finland) spectrograph, working in the near infrared wavelength range (1000–1700 nm), with a spectral resolution of 5 nm.

The imaging spectrograph is coupled with a temperature-stabilized InGaAs photo-diode array (320 × 240 pixels in image frame), positioned above a light source. The light source consists of a diffused light cylinder architecture that embeds five halogen bulbs. This architecture is mounted on a conveyor belt, 26 cm wide and 160 cm long, and is able to move at a variable speed of up to 50 mm/s (DV S.R.L., Padova, Italy). The entire system is controlled by a personal computer.

The spectrograph ImSpector$^{TM}$ N17E was calibrated by acquiring a dark image $D_i$ with the camera lens completely closed and by measuring a white reference image ($W_i$) on a standardized white Spectralon$^{®}$ ceramic material. The reflectance image ($R_i$) is then computed using the collected spectra image ($R_{0i}$) by the ratio $(R_{0i} - D_i)/(W_i - D_i)$. The calibration procedure was performed using the Spectral Scanner software (Version 1.2; SPECIM Ltd., Oulu, Finland). The same software was used to acquire and collect hyperspectral data. Hyperspectral images were then analyzed using PLS_Toolbox (Version 8.7; Eigenvector Research, Inc., Wenatchee, WA, USA) and MIA_Toolbox (Version 3.0; Eigenvector Research, Inc., Wenatchee, WA, USA) under MATLAB (Version R2019a, The Mathworks, Inc., Natick, MA, USA) environment.

### 2.2.2. Portable Spectrophotoradiometer

The ASD FieldSpec® 4 Standard–Res (Malvern Panalytical—Spectris Company, London, UK) field portable spectrophotoradiometer was used for acquiring spectra in reflectance mode. This portable instrument is able to acquire spectra in the Vis–SWIR regions (350–2500 nm) with a spectral resolution of 3 nm at 700 nm, and 10 nm at 1400/2100 nm [32].

A detector unit and a fiber optics cable connected to a contact probe, controlled by a personal computer, compose the spectroradiometer. The detection unit is realized by coupling different separate holographic diffraction gratings with three separate detectors. The detection architecture consists of a VNIR detector (512 element silicon array: 350–1000 nm), a SWIR 1 detector (Graded Index InGaAs. Photodiode, Two Stage TE Cooled; 1001–1800 nm), and a SWIR 2 detector (Graded Index InGaAs. Photodiode, Two Stage TE Cooled; 1801–2500 nm).

The ASD Contact Probe for reflectance measurements is made up of a halogen bulb light source with a color temperature of 2901 +/− 10% °K and its spot size is 10 mm. Data acquisition and calibration procedures were carried out through RS$^3$ software (Version 6.02—Malvern Panalytical—Spectris Company, London, UK) [33].

The spectroradiometer calibration was performed by referencing the dark current calibration file and by means of a white reference measurement, acquiring a standardized white Spectralon® ceramic material. After this calibration stage, the spectrum is acquired, and reflectance is then computed for each sample.

The spectroradiometer spectra ".asd" data files were stacked into an ASCII text file using ViewSpec Pro Ver. 6.2.0. The ASCII text files were then imported into MATLAB® environment (MATLAB R2019a; The Mathworks, Inc., Natick, MA, USA) using an ad hoc written routine.

Imported data files were analyzed using Eigenvector Research, Inc PLS_toolbox (Version 8.2, Eigenvector Research, Inc., Wenatchee, WA, USA) running in MATLAB® environment. Data was saved into dataset objects (DSO), and classes were assigned.

### 2.3. Spectral Data Collection, Processing and Analysis

### 2.3.1. Experimental Procedure

Three experimental setups were considered:

- 1st experimental setup. Identification of the following three classes of products: 100% cotton, 100% viscose and a blend of them (20% Cotton–80% viscose);
- 2nd experimental setup. Recognition of 100% cotton, 100% silk and a blend consisting in 50% cotton and 50% silk;
- 3rd experimental setup. Recognition of 100% cotton, 100% silk, 100% viscose and their blends (i.e., 20% Cotton–80% viscose and 50% Cotton–50% silk).

Chemometric methods were applied to recognize the studied fabrics.

### 2.3.2. Data Handling and Explorative Analysis

Acquired fabric sample images were split into two portions: a training set and a validation set. The first one was used to calibrate the classification model, whereas the second one tested the ability to discriminate between the different textile classes. In more detail, the Regions of Interest (ROIs) used to create the training and validation images are reported in Figure 2.

For each fabric sample, five random spectra were acquired on their surface in reflectance mode with the ASD portable spectrophotoradiometer, for a total of twenty-five collected spectra. Collected reflectance spectra were cut to study only the NIR range (i.e., 1000–1650 nm).

Different pre-processing algorithms were utilized in order to highlight the spectral differences between the investigated textile classes: Detrend, Savitzky-Golay (S-G) Smoothing filter, Mean Center (MC), and their combinations.

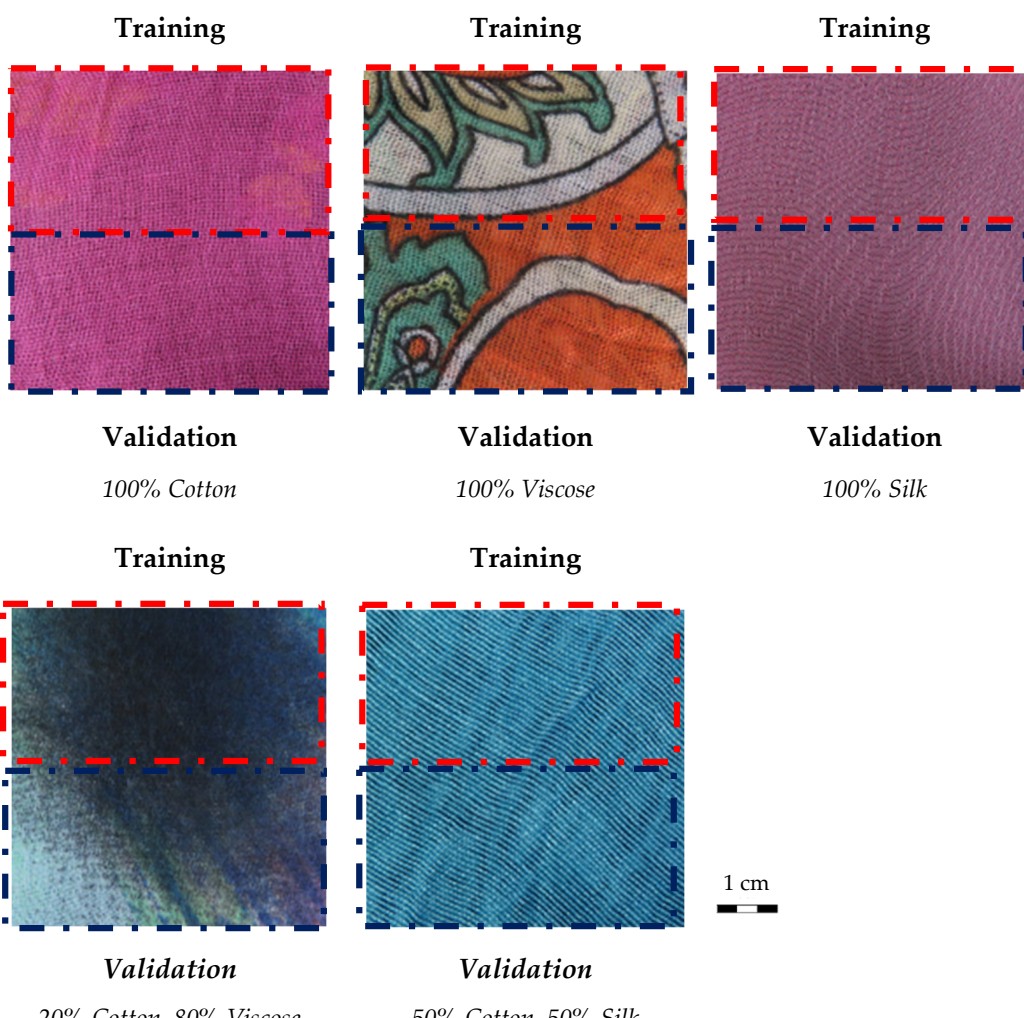

**Figure 2.** Regions of Interest (ROIs) used to create the training (red dotted area) and the validation (blue dotted area) images for each experimental set-up.

In more detail, the Detrend algorithm was applied to remove constant, linear, or curved offset [34]. (S-G) Smoothing filter was used for reducing high frequency noise in the spectral data [35]. Finally, MC, one of the most common preprocessing methods, was used to remove constant offset, which is not interesting for data variance interpretation [36].

Furthermore, Principal Component Analysis (PCA) was used to explore the collected data, followed by Partial Least Squares Discriminant Analysis (PLS-DA) classification method. PCA is an unsupervised method that allows for the dimensionality reduction in the considered spectral data matrix, which contains multiple interrelated variables, while retaining as much variation as possible [37]. The processed spectral data are decomposed into several principal components (PCs), which are linear combinations of the data, embedding the spectral variations. The first few PCs produced by the PCA are commonly used to analyze similar features among samples. In fact, in the score plots of the first two or three principal components, spectra with similar shape tend to aggregate.

PCA was applied to the different training sets of the three experimental setups for exploring the variability of spectral data and to set classes for the further classification step.

### 2.3.3. Classification Procedure

To reach the classification target, Partial Least Squares Discriminant Analysis (PLS-DA) was applied. PLS-DA is a supervised classification method, combining the properties of partial least squares regression with the distinguishing ability of a classification technique [38].

This technique is used to predict known classes in an unknown data set, and it requires prior knowledge of the data, thus known samples are used to build the classification model. PLS-DA was applied to each experimental setup in order to perform textile sample classification. Models were calibrated using the spectral information contained in the training sets and cross-validated to assess the optimal complexity of the models. PLS-DA models were built by adopting the same pre-treatment algorithms used to perform the previous explorative analysis. The built PLS-DA models were then tested on validation images of the different textile fabrics.

In order to evaluate the classifier performances, *Sensitivity*, *Specificity*, *Precision* (*P*), and Error rate (*Err*) statistical parameters were calculated [39], according to the following equations:

$$Sensitivity = \frac{TP}{TP + FN} \tag{1}$$

$$Specificity = \frac{TN}{FP + TN} \tag{2}$$

$$Precision = \frac{TP}{TP + FP} \tag{3}$$

$$Err = 1 - Precision \tag{4}$$

where: *TP* (*True Positive*) is a positive instance that is classified as positive; *FN* (*False Negative*) is a positive instance that is classified as negative; *TN* (*True Negative*) is a negative instance that is a classified as negative; and *FP* (*False Positive*) is a negative instance that is classified as positive.

## 3. Results and Discussion

### 3.1. Hyperspectral Imaging

The acquired raw and preprocessed spectra are reported in Figure 3. The absorption bands in the NIR field are mainly due to the stretching vibration of hydrogen groups such as N–H, O–H, and C–H [40]. In more detail, cotton fibers are mainly made of cellulose, including C–H, C–C and O–H groups. The absorption showed around 1480 nm is related to the first overtone of the O–H stretching from semicrystalline cellulose, characterizing cotton fibers [41]. The characteristic bands of silk, between 1540 nm and 1580 nm, related to the NH groups, are also visible in the analyzed silk and silk blend. On the contrary, the cotton band at about 1480 nm is present in the analyzed cotton-silk blend, but it is not so evident in the studied cotton-viscose blend. This is probably due to the influence of viscose that is present in a higher percentage (i.e., 80% for viscose vs. 20% for cotton). Viscose absorption bands between 1100 nm and 1200 nm, and between 1350 nm and 1400 nm, linked to the CH groups, are, in fact, preserved in the cotton-viscose blend NIR spectrum.

The analysis of the score plot allows us to identify pixel groupings, for each experimental set up, according to their spectral signature (Figure 4). Most of the variance is captured by the first two PCs. More in detail, PC1 explains 96.88% of the variance in the 1st Experimental setup, 87.74% in the 2nd Experimental setup, and 87.12% in the 3rd Experimental setup. Furthermore, PC2 explains 1.24% of the variance in the 1st Experimental setup, 7.24% in the 2nd Experimental setup, and 9.61% in the 3rd Experimental setup.

As shown in Figure 4a, spectral data is clustered into three distinct groups according to their spectral signatures: PC1 discriminates "100% Cotton" from the other two textile types, whereas "100% Viscose" and "20% Cotton–80% Viscose" are partially overlapped in the positive space of PC1. Regarding the 2nd Experimental setup (Figure 4b), PC1 separates "100% Silk" from the other two categories, whereas PC2 sharply separates "100% Cotton" from "50% Cotton–50% Silk". In the 3rd Experimental setup, a more complex scenario appears: PC2 allows us to discriminate "100% Silk" from the other samples. The "100% Viscose" scores are separated from the other sample scores, but they are nearer to the scores of the "20% Cotton–80% Viscose" class.

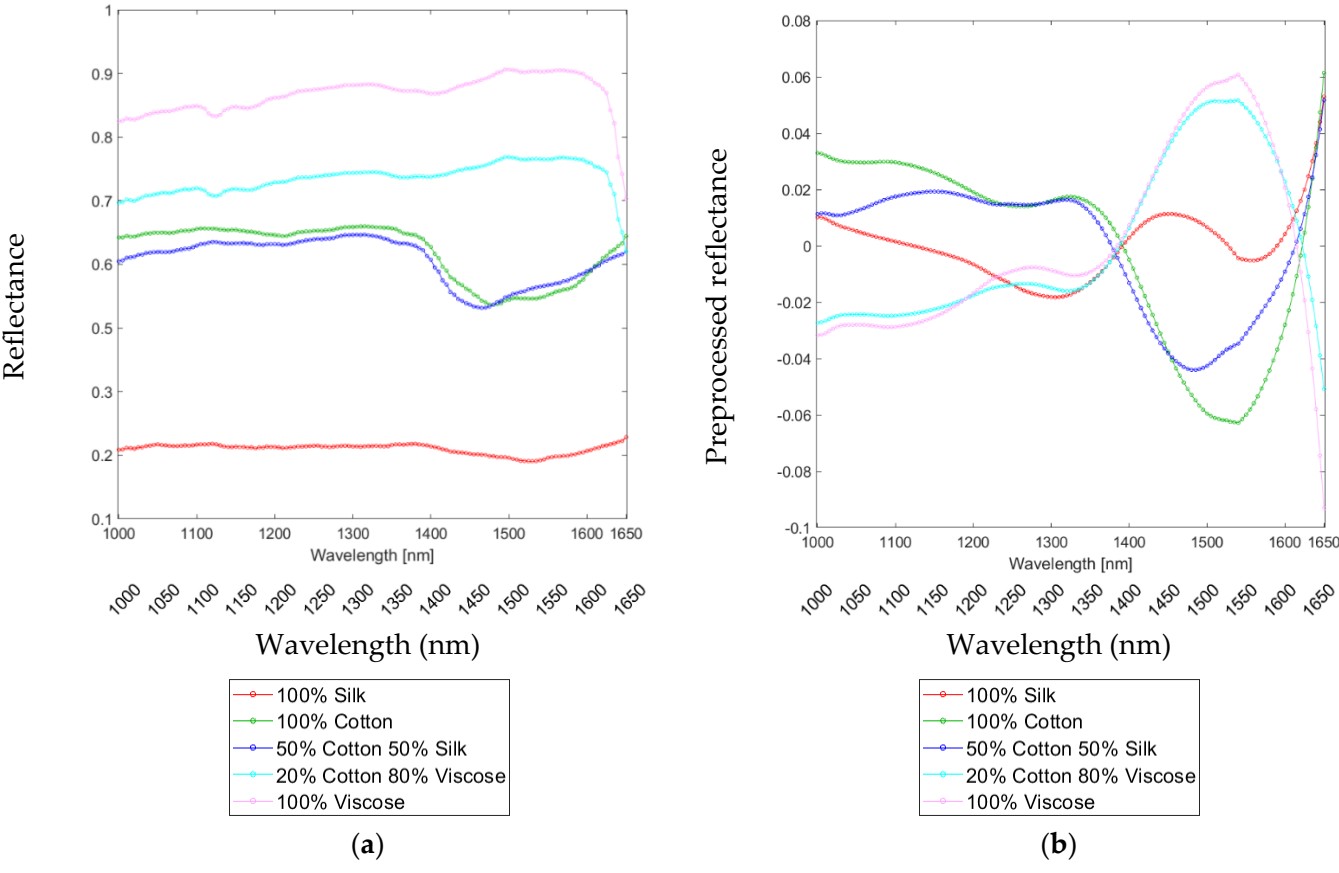

**Figure 3.** Raw (**a**) and preprocessed spectra (**b**) after the application of Detrend, Smoothing, and Mean Center algorithms of "100% Cotton", "50% Cotton–50% Silk", "20% Cotton–80% Viscose", "100% Viscose", and "100% Silk" samples.

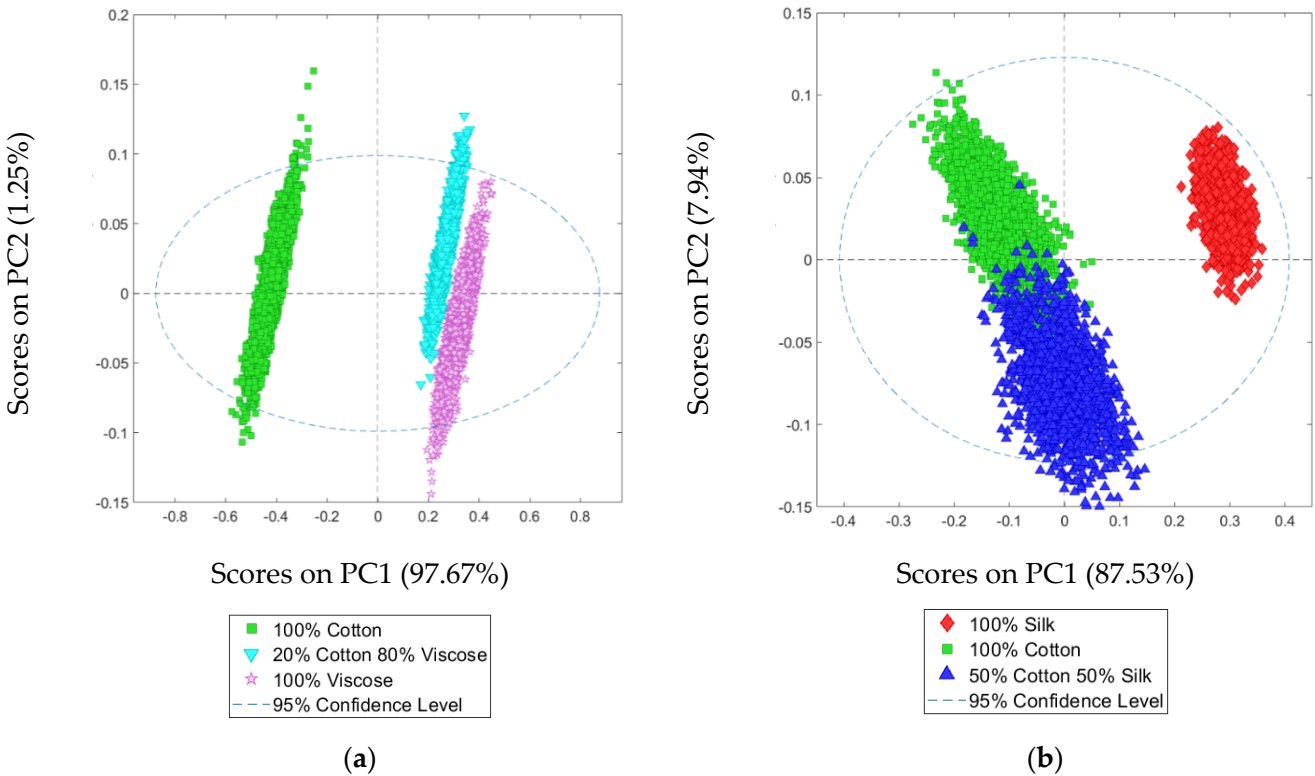

**Figure 4.** *Cont*.

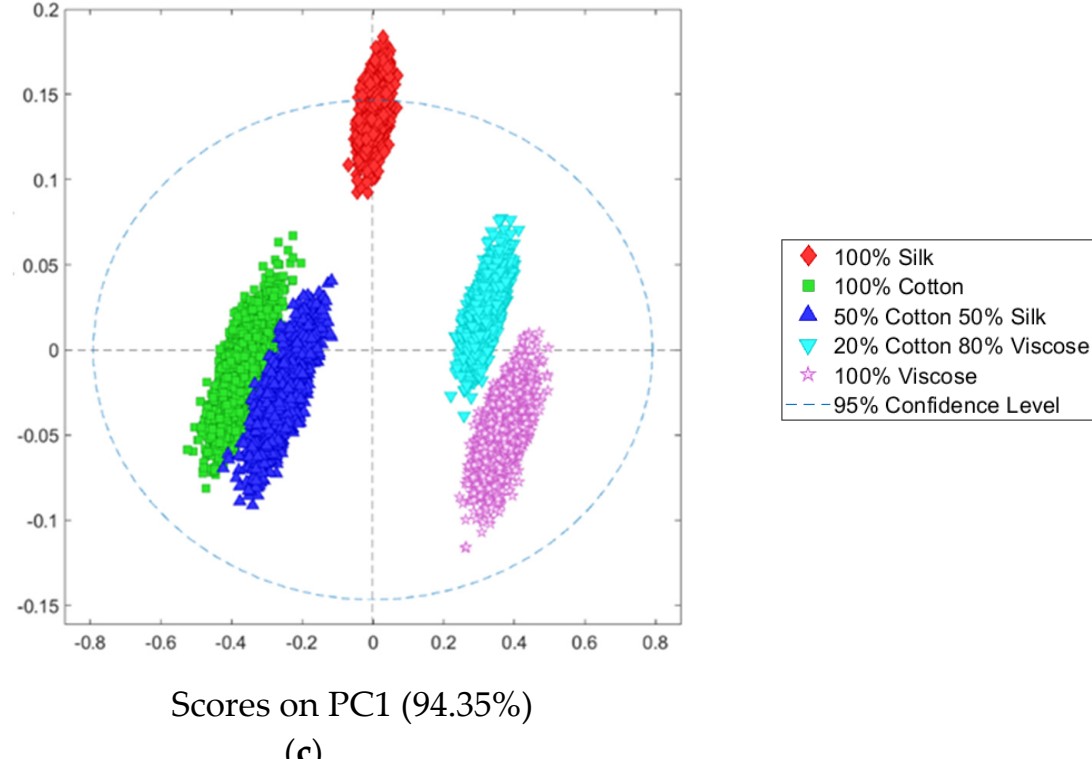

(**c**)

**Figure 4.** PCA score plot (PC1–PC2) for the 1st Experimental setup (**a**), the 2nd Experimental setup (**b**), and the 3rd Experimental set up (**c**) of the HSI average spectral signature in the NIR wavelength region (1000–1700 nm).

The obtained PLS-DA results, in terms of classification images, are shown in Figures 5–7, while the corresponding performance parameters are reported in Tables 1–3.

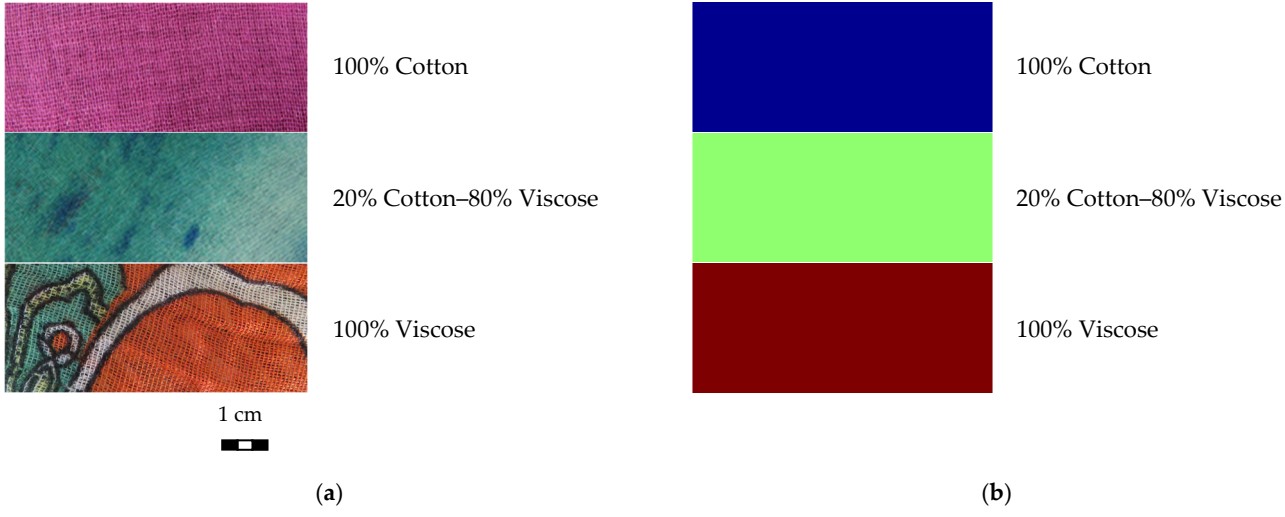

(**a**)                                                                         (**b**)

**Figure 5.** Digital image (**a**) and predicted image (**b**) of the three different analyzed textile types (i.e., "100% Cotton", "20% Cotton–80% Viscose 80%", and "100% Viscose") corresponding to the 1st Experimental setup.

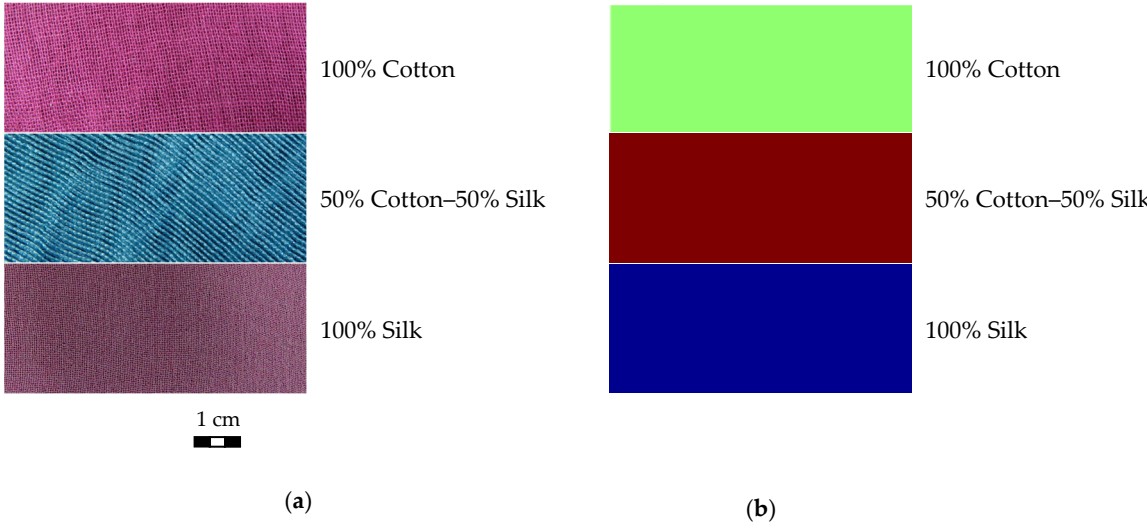

**Figure 6.** Digital image (**a**) and predicted image (**b**) of the three different analyzed textile types (i.e., "100% Cotton", "50% Cotton 50%-Silk", and "100% Silk") corresponding to the 2nd Experimental setup.

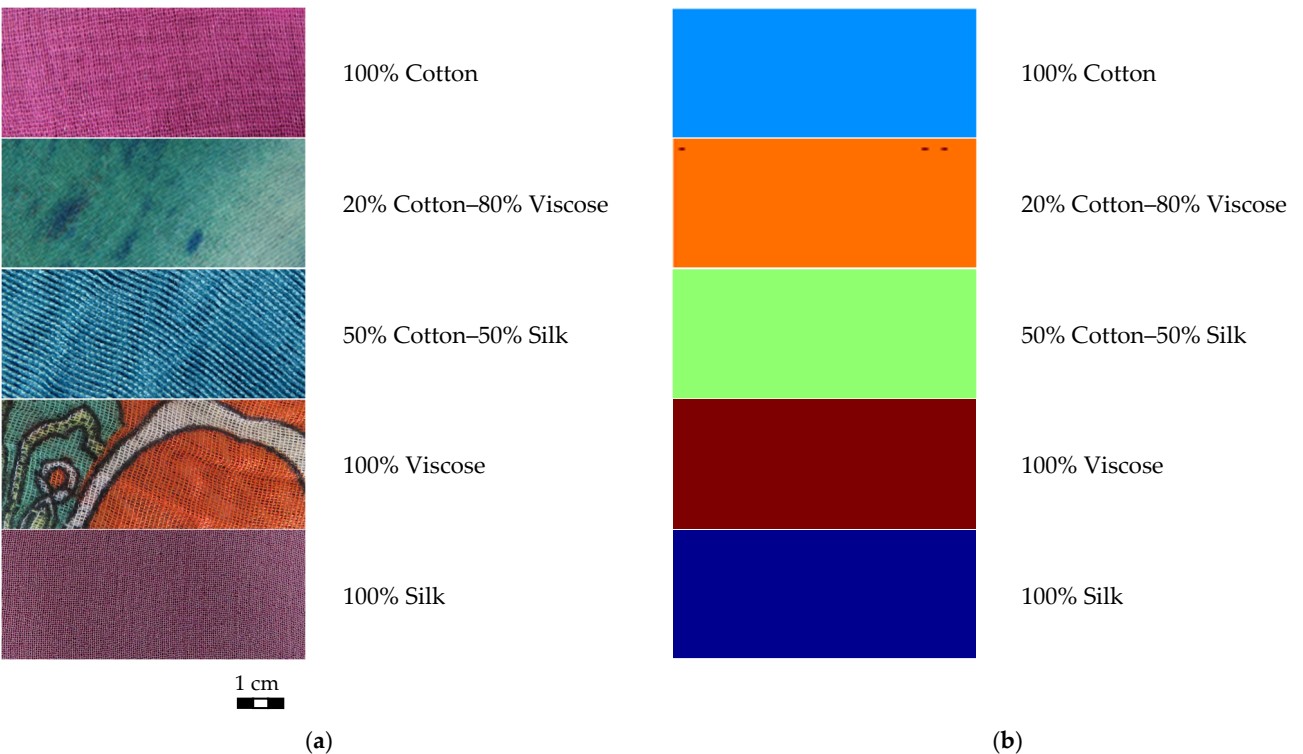

**Figure 7.** Digital image (**a**) and predicted image (**b**) of the five different analyzed textile types (i.e., "100% Cotton", "100% Viscose", "50% Cotton–50% Silk", "20% Cotton–80% Viscose", and "100% Silk") corresponding to the 3rd Experimental setup.

**Table 1.** Performance indicators (prediction results) for PLS-DA classification model, based on HSI acquisition, referred to "100% Cotton", "20% Cotton–80% Viscose", and "100% Viscose" samples, corresponding to the 1st Experimental setup.

| Class | *Sensitivity* | *Specificity* | *Err* | *P* |
|---|---|---|---|---|
| Cotton 100% | 1.000 | 1.000 | 0.000 | 1.000 |
| Cotton 20%–Viscose 80% | 0.997 | 0.999 | 0.001 | 0.999 |
| Viscose 100% | 0.999 | 0.999 | 0.001 | 0.998 |

**Table 2.** Performance indicators (prediction results) for PLS-DA classification model, based on HSI acquisition, referred to "100% Cotton", "50% Cotton–50% Silk", and "100% Silk" samples, corresponding to the 2nd Experimental setup.

| Class | *Sensitivity* | *Specificity* | *Err* | *P* |
|---|---|---|---|---|
| 100% Cotton | 1.000 | 1.000 | 0.000 | 1.000 |
| 50% Cotton–50% Silk | 1.000 | 1.000 | 0.000 | 1.000 |
| 100% Silk | 1.000 | 1.000 | 0.000 | 1.000 |

**Table 3.** Performance indicators (prediction results) for PLS-DA classification model, based on HSI acquisition, referred to "100% Cotton", "100% Viscose", "100% Silk", "20% Cotton–80% Viscose", and "50% Cotton–50% Silk" samples, corresponding to the 3rd Experimental setup.

| Class | *Sensitivity* | *Specificity* | *Err* | *P* |
|---|---|---|---|---|
| 100% Silk | 1.000 | 1.000 | 0.000 | 1.000 |
| 20% Cotton–80% Viscose | 0.990 | 0.992 | 0.008 | 0.992 |
| 50% Cotton–Silk 50% | 1.000 | 1.000 | 0.000 | 1.000 |
| 100% Viscose | 0.975 | 0.997 | 0.008 | 0.997 |
| 100% Cotton | 1.000 | 1.000 | 0.000 | 1.000 |

PLS-DA allowed the correct identification of the different categories. Only some pixels were misclassified between "100% Viscose" and "20% Cotton–80% Viscose". Indeed, the *Precision* related to these two categories is nearly 99% for both the experimental setups in which "100% Viscose" and "20% Cotton–80% Viscose" are present. The spectral signature of the two textile types is very similar since the viscose content in the "20% Cotton–80% Viscose" blend is very high. The performances of the classification models are excellent and very promising, since *Specificity* values range from 0.997 to 1.000, *Sensitivity* values range from 0.975 to 1.000, and *Precision* (*P*) values range from 0.992 to 1.000. Moreover, the Error rate (*Err*) is very low, ranging from 0 to 0.001.

### 3.2. Single-Spot Spectra

Raw and preprocessed spectra are reported in Figure 8, whereas the obtained PCA outputs in terms of score plots are reported in Figure 9. In the 1st Experimental setup, PC1 and PC2 explain 97.74% and 2.23% of the variance, respectively, whereas in the 2nd Experimental setup, PC1 explains 96.29% of the variance and PC2, 3.65%. In the 3rd Experimental setup, PC1 captures 96.29% of the variance and PC2, 3.65%. Additionally, in this case, it is possible to note the grouping of the scores based on the textile types. In the 1st Experimental setup, PC1 allows us to distinguish "100% Viscose" from "100% Cotton", whereas PC2 discriminates the cotton-viscose blend from the "pure" fibers. In the 2nd Experimental setup, PC1 separates "100% Silk" from the others, whereas PC2 allows for discerning between "100% Cotton" and "50% Cotton–50% Silk". In the 3rd Experimental setup, sample grouping according to the textile fiber classes is easily detectable and the situation is very similar to that observed in the corresponding Experimental setup carried out by the hyperspectral imaging approach.

In Tables 4–6, the prediction results in terms of *Sensitivity*, *Specificity*, Error rate (*Err*), and *Precision* (*P*) for each experimental setup are shown.

PLSDA classification methods enable accurate single-spot spectrum identification since *Sensitivity*, *Specificity*, and *Precision* are always 1 while the Error rate is 0.

Single spectra classification results are not influenced by sample surface characteristics (i.e., roughness, crease presence, etc.) and light scattering phenomena because the simpler acquisition process allows for more homogeneous conditions than those of the HSI scanner.

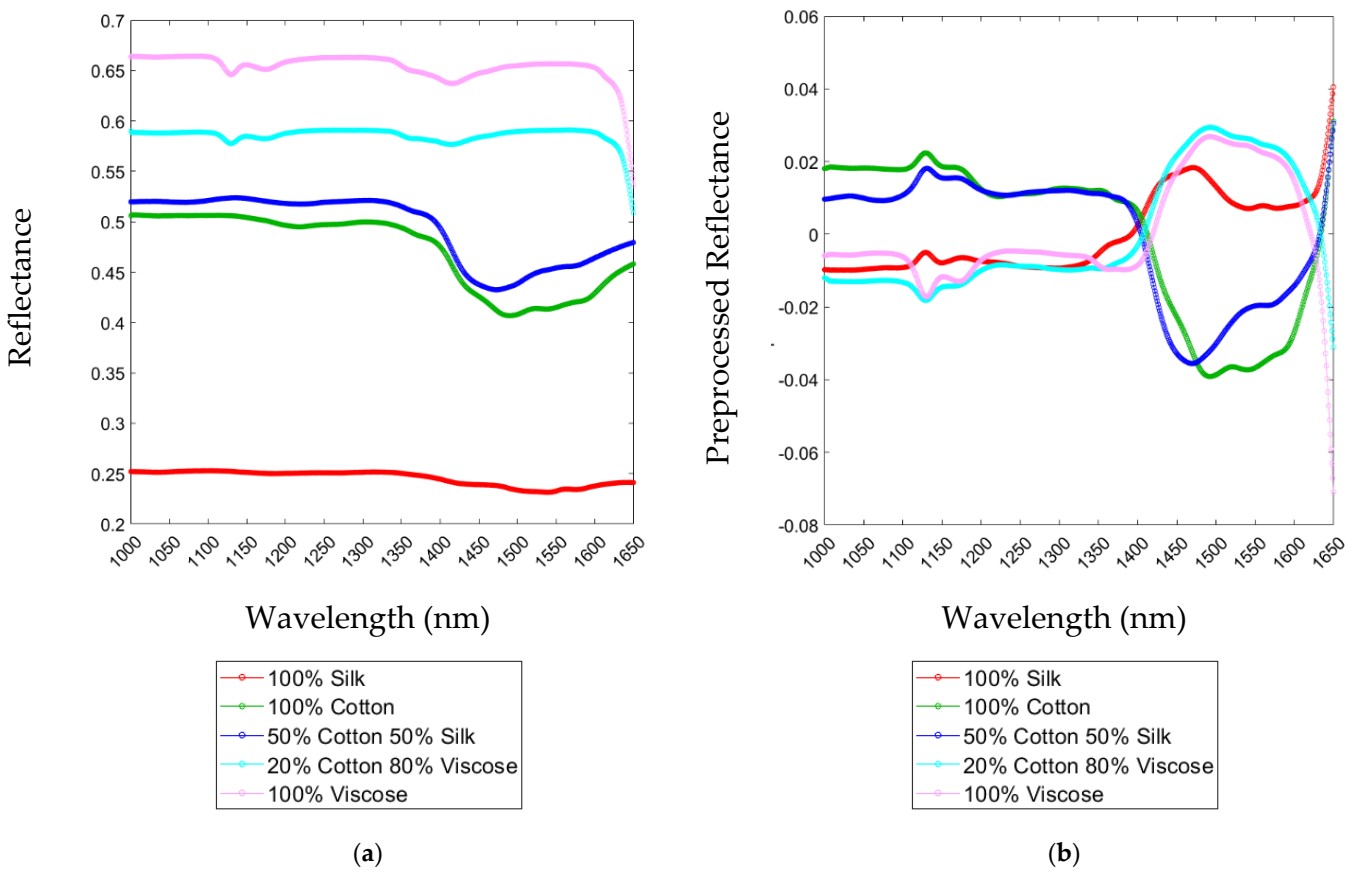

**Figure 8.** Raw (**a**) and preprocessed spectra (**b**) after applying Detrend, Smoothing, and Mean Center algorithms of "100% Cotton", "50% Cotton–50% Silk", "20% Cotton–80% Viscose", "100% Viscose", and "100% Silk" samples.

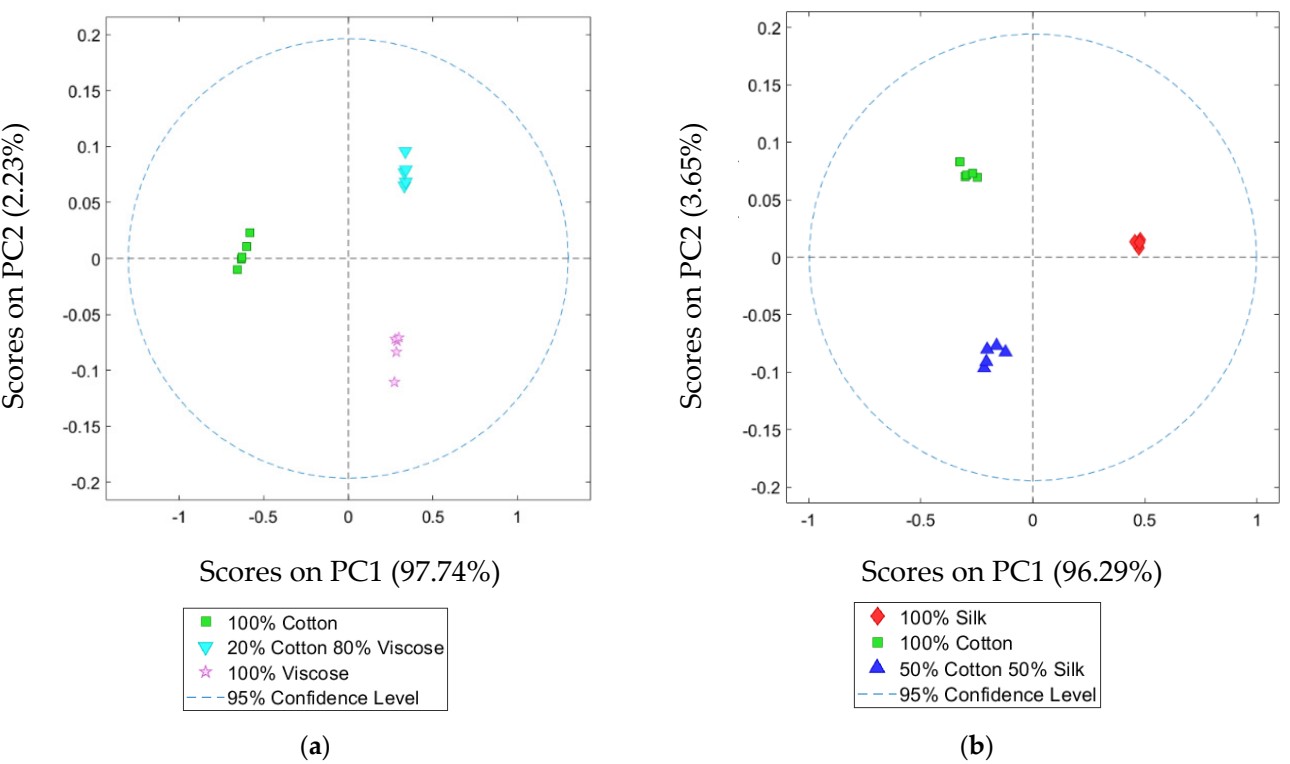

**Figure 9.** *Cont*.

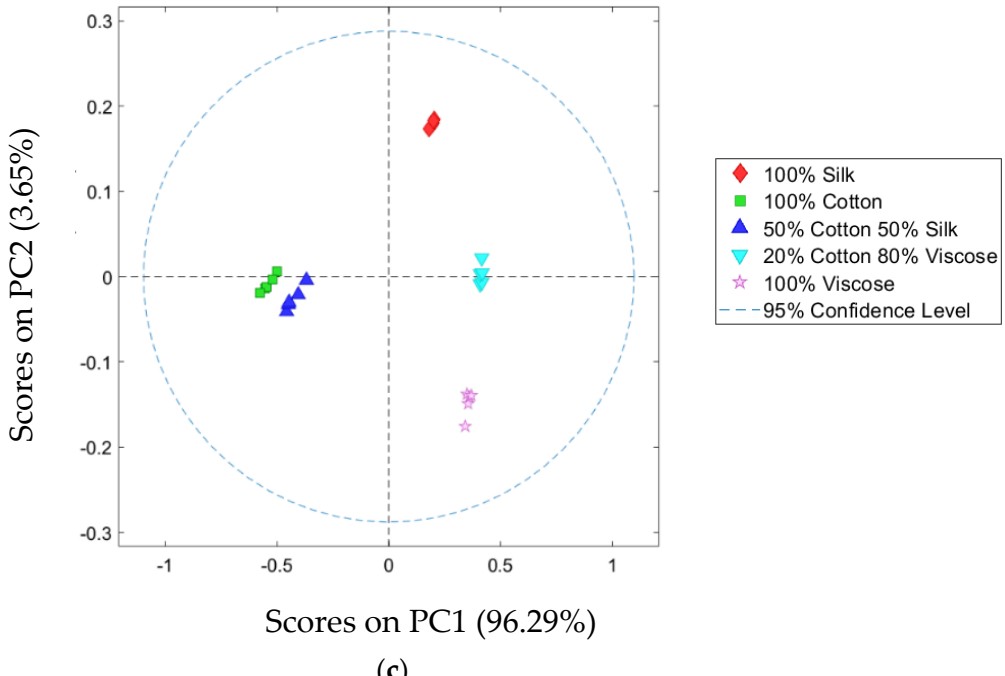

**(c)**

**Figure 9.** PCA score plot (PC1–PC2) for the 1st Experimental setup (**a**), the 2nd Experimental setup (**b**), and the 3rd Experimental setup (**c**) on single-spot spectra in the NIR wavelength region (1000–1700 nm).

**Table 4.** Performance indicators (prediction results) for PLS-DA classification model to perform the classification based on single-spot collected spectra of "100% Cotton", "20% Cotton–80% Viscose", and "100% Viscose" sample types corresponding to the 1st Experimental setup.

| Class | Sensitivity | Specificity | Err | P |
|---|---|---|---|---|
| 100% Cotton | 1.00 | 1.00 | 0.00 | 1.00 |
| 20% Cotton–80% Viscose | 1.00 | 1.00 | 0.00 | 1.00 |
| 100% Viscose | 1.00 | 1.00 | 0.00 | 1.00 |

**Table 5.** Performance indicators (prediction results) for PLS-DA classification model to perform the classification based on single-spot collected spectra of "100% Cotton", "50% Cotton–50% Silk", and "100% Silk" sample types corresponding to the 2nd Experimental setup.

| Class | Sensitivity | Specificity | Err | P |
|---|---|---|---|---|
| 100% Cotton | 1.00 | 1.00 | 0.00 | 1.00 |
| 50% Cotton–50% Silk | 1.00 | 1.00 | 0.00 | 1.00 |
| 100% Silk | 1.00 | 1.00 | 0.00 | 1.00 |

**Table 6.** Performance indicators (prediction results) for PLS-DA classification model to perform the classification based on single-spot collected spectra "100% Cotton", "100% Viscose", "100% Silk", "20% Cotton–80% Viscose", and "50% Silk–50% Cotton" sample types corresponding to the 3rd Experimental setup.

| Class | Sensitivity | Specificity | Err | P |
|---|---|---|---|---|
| 100% Silk | 1.00 | 1.00 | 0.00 | 1.00 |
| 20% Cotton–80% Viscose | 1.00 | 1.00 | 0.00 | 1.00 |
| 50% Cotton–50% Silk | 1.00 | 1.00 | 0.00 | 1.00 |
| 100% Viscose | 1.00 | 1.00 | 0.00 | 1.00 |
| 100% Cotton | 1.00 | 1.00 | 0.00 | 1.00 |

## 4. Conclusions and Future Perspectives

This work was carried out in order to verify the possibility of applying NIR spectroscopy to end-of-life textile recycling in agreement with the principles of circular economy. This goal was achieved using two different devices: a HSI system and a portable spectrophotoradiometer. The obtained results show that the applied techniques allow us to correctly identify the different textile types, reaching a *Precision* rate greater than 99.2% for HSI images and around 100% for single-spot data. Useful information about the composition of textile waste types were thus obtained, allowing us to classify pure fabric (i.e., "100% Cotton", "100% Silk", and "100% Viscose") and some blends (i.e., "20% Cotton–80% Viscose" and "50% Cotton–50% Silk").

The two different NIR spectra acquisition techniques can be used individually or in a complementary way. The portable instrument could be applied to perform a rapid test on textile waste that is fed to the recycling plant, on the recovered final products, and/or on samples collected at different operation stages for process control. In addition, the HSI system can be utilized not only for the same purposes, but also as a sensor-based sorting system.

The developed approach can be considered as a methodological procedure to be systematically implemented at a recycling plant scale. In fact, by applying these analytical methods, the recycling process could be efficiently automatized, complementing or replacing manual operations/sorting stages. In this way, recycling operations could be sped up and improved, thus realizing a significant cost reduction. Moreover, a more circular and sustainable system would contribute to achieve many of the UN Sustainable Development Goals (SDGs). Indeed, for the textile industry, "SDG 12: Responsible Consumption and Production" is a gateway to many other SDGs, including "SDG 6: Clean water and Sanitation", "SDG 7: Affordable and Clean Energy", and "SDG 13: Climate Action".

Starting from the obtained results, further classification models will be developed and applied on a greater variety of textile waste samples, both pure fabrics and blends, not only to classify as many fabrics as possible, but also to quantify their content in terms of fabric type percentage.

**Author Contributions:** Conceptualization, G.B., S.S. and R.P.; methodology, R.P. and R.G.; software, R.P. and R.G.; validation, G.B., R.P., R.G. and S.S.; formal analysis, R.P. and R.G.; investigation, G.B., R.P., R.G. and S.S.; resources, G.B. and S.S.; data curation, R.P. and R.G.; writing—original draft preparation, R.P. and R.G.; writing—review and editing, G.B., R.P. and S.S.; visualization, G.B. and S.S.; supervision, G.B. and S.S. All authors have read and agreed to the published version of the manuscript.

**Funding:** This research received no external funding.

**Informed Consent Statement:** Not applicable.

**Data Availability Statement:** The data presented in this study are available on request from the corresponding authors.

**Conflicts of Interest:** The authors declare no conflict of interest.

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
