# Peer review of "End-of-Life Textile Recognition in a Circular Economy Perspective: A Methodological Approach Based on Near Infrared Spectroscopy"

_sustainability, doi:10.3390/su141610249_

Round 1

Reviewer 1 Report

The present work aims to explore the possibility of using near infrared spectroscopy to recognize textile fibers, a hyperspectral imaging system and a portable single point spectroradiometer, has been developed, implemented and tested for textile fiber sorting and/or quality control system in a recycling process.

The enormous environmental and social challenges have made traceability a key priority in the textile industry due to the increasing pressure for accountability and transparency. At the same time, traceability has also been identified as a key lever to enhance circularity and thus reduce the sector's footprint. However, extending traceability for this purpose is particularly challenging for textile companies, due to the multiplicity of actors and aspects to be considered. The literature on the subject is particularly confusing and fragmented, with no framework for understanding its systematic implementation throughout the life cycle. 

The research is mainly focused on upstream traceability of raw materials and manufacturing stages and there is a clear lack regarding the implementation of traceability for circularity applications beyond the distribution stage. This article opens up important research avenues and beneficial perspectives for practitioners.

Author Response

The authors are grateful for the words of encouragement written by the reviewer. Furthermore, we fully agree on the actual lack, underlined by Reviewer 1, “regarding the implementation of traceability for circularity applications beyond the distribution stage”. The approach we proposed could represent an important tool to specifically develop and implement traceability logics in all the different textiles production and manipulation stages.

Reviewer 2 Report

Comments:

In this article, Giuseppe Bonifazi et al. reported a novel method for identifying textiles by using near-infrared spectroscopy. However, some issues should be addressed before published:

1. Please list the calculation process of scores on PC1-2 in detail.

2. The practicability of proposed method should be checked through series unknown fabrics, and the results should be compared with stander methods.

3. The language of this article should be checked carefully.

Author Response

Thank you for the constructive comments, suggestions and recommendations. We found them very valuable and we believe that they have improved the overall quality of the paper.

  1. Please list the calculation process of scores on PC1-2 in detail.

Thank you for your comment. We have now extended the Principal Component Analysis (PCA) explanation in the paragraph ‘2.3.2. Data handling and explorative analysis’, page 5, lines 213-221: “PCA is an unsupervised method that allows the dimensionality reduction of the considered spectral data matrix, which contains multiple interrelated variables, while retaining as much variation as possible [37]. The processed spectral data are decomposed into several principal components (PCs), which are linear combinations of the data embedding the spectral variations. The first few PCs produced by PCA are commonly used to analyze similar features among samples. In fact, in the score plots of the first two or three principal components, spectra with similar shape tend to aggregate.”

  1. The practicability of proposed method should be checked through series unknown fabrics, and the results should be compared with stander methods.

Thanks for the suggestion which we will certainly take into account for our future studies. We are planning to test unknown samples on a new model made up of a greater variety of fabrics. The aim of the current work is to propose a methodological approach, and not a systematic study, for the classification of textile by NIR spectroscopy, as also indicated in the manuscript title. In this perspective, we believe that the investigated samples are sufficient to demonstrate the validity of the proposed approach in a “methodological perspective”.

  1. The language of this article should be checked carefully.

Thank you for your suggestion. We have revised the language throughout the manuscript.